# Optogenetic control of cellular forces and mechanotransduction

Léo Valon[1], Ariadna Marín-Llauradó[1], Thomas Wyatt[2,3], Guillaume Charras[3,4] & Xavier Trepat[1,5,6,7]

Contractile forces are the end effectors of cell migration, division, morphogenesis, wound healing and cancer invasion. Here we report optogenetic tools to upregulate and down-regulate such forces with high spatiotemporal accuracy. The technology relies on controlling the subcellular activation of RhoA using the CRY2/CIBN light-gated dimerizer system. We fused the catalytic domain (DHPH domain) of the RhoA activator ARHGEF11 to CRY2-mCherry (optoGEF-RhoA) and engineered its binding partner CIBN to bind either to the plasma membrane or to the mitochondrial membrane. Translocation of optoGEF-RhoA to the plasma membrane causes a rapid and local increase in cellular traction, intercellular tension and tissue compaction. By contrast, translocation of optoGEF-RhoA to mitochondria results in opposite changes in these physical properties. Cellular changes in contractility are paralleled by modifications in the nuclear localization of the transcriptional regulator YAP, thus showing the ability of our approach to control mechanotransductory signalling pathways in time and space.

[1] Institute for Bioengineering of Catalonia, Barcelona 08028, Spain. [2] MRC Laboratory for Molecular Cell Biology, University College London, Gower Street, London WC1E 6BT, UK. [3] London Centre for Nanotechnology, London WC1H 0AH, UK. [4] Department of Cell and Developmental Biology and Institute for the Physics of Living Systems, University College London, London WC1E 6BT, UK. [5] Facultat de Medicina, Universitat de Barcelona, 08036 Barcelona, Spain. [6] Institució Catalana de Recerca i Estudis Avançats (ICREA), 08010 Barcelona, Spain. [7] Centro de Investigación Biomédica en Red en Bioingeniería, Biomateriales y Nanomedicina, Barcelona 08028, Spain. Correspondence and requests for materials should be addressed to X.T. (email: xtrepat@ibecbarcelona.eu).

A broad variety of biological processes in development, homeostasis and disease are driven by mechanical forces generated by the contractile actomyosin cytoskeleton. During the course of morphogenesis, these forces are tightly regulated to drive tissue elongation, invagination, branching and vascularization[1,2]. Contractile forces also control key steps in wound healing, including angiogenesis, re-epithelialization and remodelling of the newly synthesized connective tissue[3,4]. Aberrant contractility of the smooth muscle and endothelium underlies pathological processes such as bronchospasm in asthma and vasoconstriction in arterial hypertension[5,6]. In cancer, contractile forces drive diverse aspects of invasion and metastasis, from propulsion of cell migration to remodelling of the extracellular matrix by cancer cells and stromal fibroblasts[7–9]. At the subcellular level, contractile forces enable cell adhesion, polarization, division and mechanosensing[10–14]. In all these physiological and pathological processes, physical forces are tightly regulated—or altogether deregulated—in space and time.

The central role of contractile forces in cell function has motivated extensive research to identify the underlying molecular mechanisms and regulatory pathways. From this fundamental knowledge several chemical compounds have been developed to tune cellular force generation. Some of these compounds, such as bronchodilators and vasodilators that act on smooth muscle cells, are routinely used in disease management[15–17], while others are restricted to basic research. A common strategy to target cell contractility is to use small molecules acting directly on the motor domain of myosin II, such as blebbistatin[18]. Alternatively, small molecules and genetic perturbations are often used to target regulatory pathways, such as those controlling calcium levels or Rho GTPases[19]. Despite their well-established effectiveness, the biochemical and genetic manipulations mentioned above are severely limited by their inability to provide tight spatiotemporal control of cell contractility. This impedes their use to determine how local upregulation or downregulation of contractility could lead to cellular or multicellular shape changes. In addition, drugs and siRNAs treatments often display poor reversibility and are prone to off-target effects.

The recent development of optogenetic technologies offers promising possibilities to control signalling pathways with high spatiotemporal resolution[20]. By expressing genetically encoded light-sensitive proteins, optogenetic technology enables the reversible perturbation of intracellular biochemistry with subcellular resolution. Optogenetics has been successfully applied to control the activity of ion channels, RhoGTPases, phospholipids, transcription factors and actin polymerization factors[21–29]. However, no previous study has established by direct measurement whether and to what extent optogenetics can be used to control cell–cell forces, cell–matrix forces and mechanotransductory signalling pathways.

Here we report two optogenetic tools based on controlling the activity of endogenous RhoA to upregulate or downregulate cell contractility. We show that these tools enable rapid, local and reversible changes in traction forces, cell–cell forces, and tissue compaction. We show, further, that changes in cellular forces are paralleled by translocation of the transcriptional regulator YAP, indicating that our tools can be used to control mechanotransductory pathways.

## Results

**Optogenetic control of RhoA activity.** RhoA is activated by several Guanine Exchange Factors (RhoA-GEFs), which localize mainly at the plasma membrane in epithelial cells. We reasoned that overexpressing the catalytic domain of a RhoA-GEF and forcing its localization to the plasma membrane should increase RhoA activity and promote cortical contractility (Fig. 1a, upper box). Conversely, forcing the localization of the same catalytic domain to mitochondria should decrease RhoA activity and relax cell contractility (Fig. 1a, lower box). To control Rho-GEF localization we used the CRY2/CIBN light-gated dimerizer system. This system is based on two proteins, CRY2 and CIBN, which bind with high affinity upon exposure to blue light, but rapidly dissociate when illumination is switched off[30].

As a candidate to control RhoA activity, we selected the DHPH domain of ARHGEF11 (refs 31,32) and fused it to CRY2-mCherry to form ARHGEF11(DHPH)-CRY2-mCherry, hereafter referred to as optoGEF-RhoA. To control the localization of this protein, we engineered two different versions of CIBN, one targeted to the plasma membrane (CIBN-GFP-CAAX) (Fig. 1b) and one targeted to the mitochondrial membrane (mito-CIBN-GFP) (Fig. 1d). To assess whether this approach enabled efficient recruitment of optoGEF-RhoA to the subcellular structures where CIBN was localized, we illuminated square areas of MDCK cells expressing either CIBN-GFP-CAAX or mito-CIBN-GFP with 488 nm light pulses (see methods). As predicted, optoGEF-RhoA was recruited to the plasma membrane in cells expressing CIBN-GFP-CAAX (Fig. 1c; Supplementary Movie 1), whereas it was recruited to mitochondria in cells expressing mito-CIBN-GFP (Fig. 1e; Supplementary Movie 2). In both cases, recruitment was limited to cells within the exposed area and, upon switching off the blue light, CRY2/CIBN complexes dissociated and the mCherry signal returned progressively to the cytoplasm. Quantitative image analysis showed that recruitment of optoGEF-RhoA to its targeted location was nearly instantaneous ($<10$ s), whereas dissociation was slower ($\sim 5$ min, Fig. 1f,g). These characteristic times are consistent with previous reports of CIBN/CRY2 kinetics[24,30]. By using an infrared RhoA biosensor consisting of the Rhotekin Binding Domain (RBD) fused to infraRed Fluorescent Protein (iRFP), we confirmed that local recruitment of optoGEF-RhoA to the cell membrane correlates with increased RhoA activity (Supplementary Fig. 1). Altogether, these experiments show that our optogenetic approach allows the catalytic domain of ARHGEF11 to be rapidly and reversibly localized to the plasma membrane or mitochondria, resulting in controlled RhoA activity.

**Optogenetic upregulation and downregulation of cell forces.** We next investigated whether RhoA activation following translocation of optoGEF-RhoA to the cell membrane or mitochondria was paralleled by changes in cell contractility. To this end, we used Traction Force Microscopy[33] to measure forces exerted by cells on their underlying soft collagen-I-coated substrate (12 kPa, polyacrylamide) during optogenetic activation and deactivation. To study the effect of optoGEF-RhoA translocation to the cell membrane, we created a MDCK cell line stably co-expressing optoGEF-RhoA and CIBN-GFP-CAAX (Fig. 2a). We then exposed selected areas of the microscope field of view to a sequence of blue light pulses (one pulse of blue light every 10 s). Exposed regions exhibited a $\sim 50\%$ increase in traction forces that tended to plateau after 1 min (Fig. 2b,g; Supplementary Movie 3). By contrast, unexposed adjacent regions exhibited no changes in tractions (Fig. 2b). Control cells stably expressing only CIBN-GFP-CAAX did not experience changes in traction forces upon illumination (Fig. 2g, black curve). To compute tension within and between cells we used Monolayer Stress Microscopy[34]. Following a behaviour similar to traction forces, cellular tension increased in exposed regions and remained unchanged in unexposed ones (Fig. 2c). Increases in traction and tension could be sustained for at least 40 min and, importantly, they were fully reversible (Fig. 2i,k). This allowed us to generate periodic

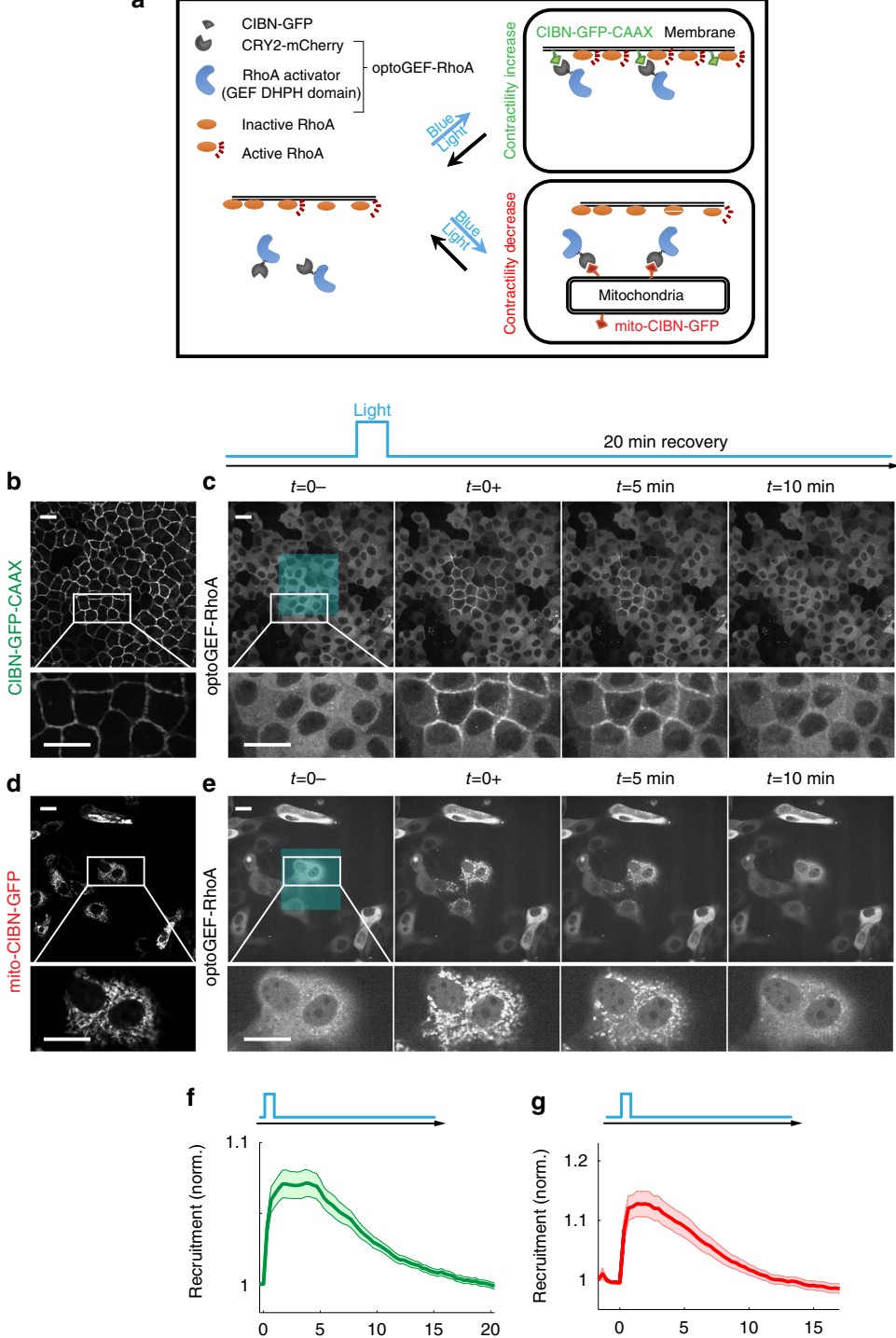

**Figure 1 | Control of optoGEF-RhoA localization. (a)** Scheme of the optogenetic system to control cell contractility. The system is based on overexpressing a RhoA activator (DHPH domain of ARHGEF11) fused to the light-sensitive protein CRY2-mcherry. The resulting protein is called optoGEF-RhoA. In the absence of blue light, a fraction of the RhoA pool is active because of endogenous activity and overexpression of the RhoA activator (left scheme). Upon illumination, CRY2 changes conformation and binds to its optogenetic partner CIBN. To increase contractility, we forced translocation of optoGEF-RhoA to the cell surface, where RhoA is located, by targeting CIBN-GFP to the plasma membrane (top right panel). To decrease contractility, we sequestered optoGEF-RhoA at mitochondria by targeting CIBN-GFP to the mitochondrial membrane (bottom right panel). **(b,c)** Confluent MDCK cells stably expressing CIBN-GFP-CAAX (**b**) and optoGEF-RhoA before and after blue light illumination (**c**). Illumination was restricted to the central area of the field of view represented by a blue square. The temporal pattern of illumination is indicated by the upper blue line. **(d,e)** Subconfluent MDCK cells co-transfected with mito-CIBN-GFP (**d**) and optoGEF-RhoA (**e**). In **b**–**e**, bottom panels show zoomed areas marked by the white rectangles. **(f,g)** Quantification of optoGEF-RhoA both at the cell membrane (**f**) and at the cell mitochondria (**g**) over the 20 min of experiment ($n = 8$ and $n = 14$ fields of view, respectively). Data are shown as mean ± s.e.m. Cells were illuminated with 2 pulses of blue light separated by 10 s at time t = 0. Scale bars, 20 μm.

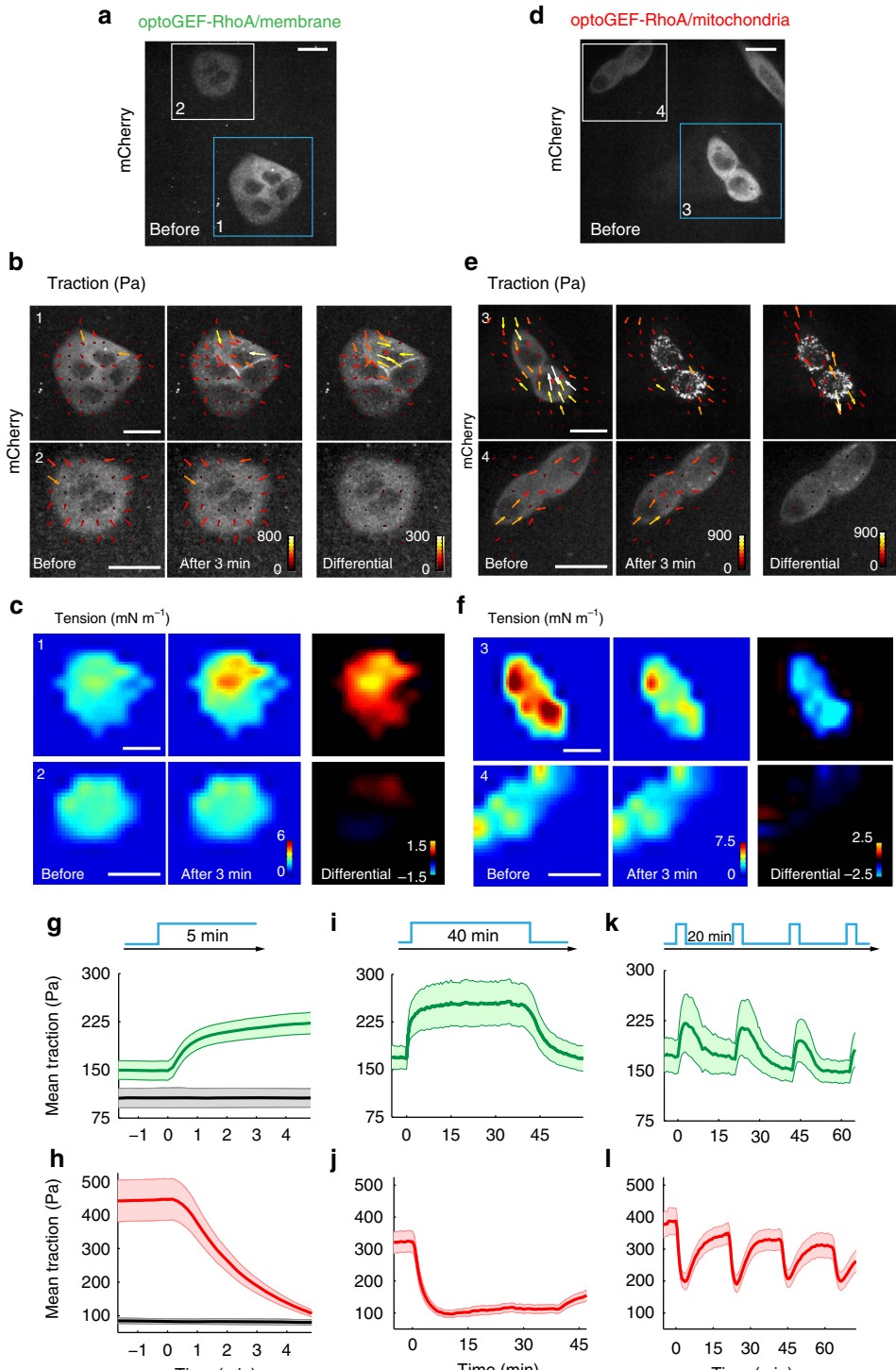

**Figure 2 | Optogenetic upregulation and downregulation of cell contractility.** (**a–f**) Cells expressing optoGEF-RhoA and either CIBN-GFP-CAAX (**a–c**) or mito-CIBN-GFP (**d–f**) were locally and transiently activated. (**a,d**) Images of optoGEF-RhoA mCherry signal one minute before optogenetic activation. At time $t = 0$, cells highlighted by blue squares (labelled 1 and 3) were illuminated, whereas cells highlighted by white squares (labelled 2 and 4) were not. (**b**) Traction forces exerted by cell clusters labelled 1 and 2 in **a** before and after optogenetic activation. (**c**) Tension within and between cells in clusters labelled 1 and 2 in **a** before and after optogenetic activation. (**e**) Traction forces exerted by clusters labelled 3 and 4 in **d** before and after optogenetic activation. (**f**) Tension within and between cells in clusters labelled 3 and 4 in **d** before and after optogenetic activation. For comparison, the vector difference of tractions and the scalar difference in tension between the two time points is shown on the right column of **b,c** and **e,f**. (**g–l**) Quantification of the mean traction amplitude over time for cells expressing CIBN-GFP-CAAX and optoGEF-RhoA (**g,i,k**) and for cells expressing optoGEF-RhoA and mito-CIBN-GFP (**h,j,l**) subjected to distinct illumination protocols. Thick lines display the means across different experiments and shaded areas indicate s.e.m. Black line in **g** corresponds to control cells expressing only CIBN-CAAX-GFP. Black line in **h** corresponds to control cells expressing mito-CIBN-GFP and CRY2-mCherry. (**g,h**) Activation for 5 min (one activation pulse every 10 s), in **g**, $n\_green = 15$ cells, $n\_black = 9$ cells, in **h**, $n\_red = 8$ cells, $n\_black = 10$ cells. (**i,j**) Activation for 40 min (one activation pulse every 30 s). In (**i**) $n = 9$ cells; in **j**, $n = 7$ cells. (**k,l**) Periodic activation separated by 20 min of recovery (with activation routines made of 2 pulses of blue light separated by 10 s). In **k**, $n = 6$ cells; in **l**, $n = 10$ cells. Scale bars, 20 μm.

and local patterns of contraction and relaxation by simply alternating periods of pulsed illumination and darkness (Fig. 2k). These patterns were reproducible in time with a modest attenuation in resting state traction.

To study the effect of optoGEF-RhoA translocation to mitochondria, we transiently co-transfected MDCK cells with mito-CIBN-GFP and optoGEF-RhoA, and subjected them to the same illumination protocols described above (Fig. 2d). Approximately 50% of the cells were successfully transfected with both plasmids. Due to overexpression of optoGEF-RhoA, baseline levels of traction forces were 4.5-fold higher than in control cells expressing mito-CIBN-GFP and CRY2-mCherry (lacking the GEF DHPH domain; Fig. 2h). Upon illumination, cell tractions and tension decreased and reached a plateau after 5 min (Fig. 2d–f,h,j; Supplementary Movie 4). By contrast, tractions and tension in unexposed adjacent cells remained constant (Fig. 2e,f). Much as in the case of cell contraction, the reversibility of cell relaxation allowed us to generate oscillatory force patterns (Fig. 2l). A stable cell line expressing mito-CIBN-GFP and optoGEF-RhoA showed similar behaviour to transiently transfected cells, the only differences being a lower traction baseline and reduced cell-to-cell variability (Supplementary Fig. 2). Altogether, these results show that controlling the subcellular localization of ARHGEF11 catalytic domain enables the spatiotemporal control of signalling and hence cell contractility.

**Optogenetic control of actin fibres and focal adhesions**. We next studied the structural cytoskeletal changes that underlie variations in forces exerted by cellular actomyosin. These may stem from generation of linear arrays of actin filaments to serve as scaffolds for myosin contractility. To follow the remodelling of actin filaments over time, we co-transfected MDCK cells with optoGEF-RhoA, either CIBN-GFP-CAAX or mito-CIBN-GFP, and lifeact-iRFP, an F-actin reporter. Illumination with blue light for 11 min showed that the optogenetic increase in contractility was paralleled by the formation of actin stress fibres (Fig. 3a,b; Supplementary Movie 5). Conversely, cell relaxation was paralleled by the disappearance of basal stress fibres. (Fig. 3c,d; Supplementary Movie 6). Both phenomena were fully reversible. Next, we examined the distribution of focal adhesions during changes of contractility by co-transfecting optogenetic constructs and vinculin-iRFP. Increasing contractility did not lead to systematic changes in focal adhesion size or distribution. However, relaxing contractility resulted in a sharp and reversible reduction of focal adhesion sites (Fig. 3e,f; Supplementary Movie 7). Thus, optogenetic contraction and relaxation correlated with structural changes in stress-generating and stress-sensitive elements of the cell.

**Optogenetic control of epithelial deformation**. Having shown the ability of our optogenetic system to generate rapid and reversible changes in cellular traction and tension, we next sought to investigate whether these changes were paralleled by tissue deformations. To this end, we first focused on confluent monolayers of a stable MDCK cell line expressing CIBN-GFP-CAAX, optoGEF-RhoA and myr-iRFP seeded on 12kPa collagen-I-coated polyacrylamide gels. myr-iRFP is a cell membrane anchor fused to iRFP that served as a reporter to quantify displacements of the lateral cell membranes during optogenetic activation. Figure 4a,b shows the overlay of two pairs of images separated by 100 s taken either before exposure (Fig. 4a) or just after illumination (Fig. 4b). Before illumination, no displacements of cell boundaries were visually discernible (Fig. 4a). Quantification with Particle Imaging Velocimetry (PIV) of myr-iRFP images showed average fluctuations of $\sim 0.5\,\mu m\,min^{-1}$ (Fig. 4c). Shortly after illumination of the central square region of the monolayer (Fig. 4b), membranes moved towards the center of the illuminated region with typical velocities of $1.5\,\mu m\,min^{-1}$, thereby indicating cell compaction (Fig. 4d; Supplementary Movie 8). Cells located immediately outside the illuminated region also moved inwards due to transmission of forces across intercellular junctions in the monolayer, but velocities vanished within $15\,\mu m$ (approximately two cell diameters). Time lapse analysis of PIV maps showed that the increase in compaction was largely restricted to the initial 40 s after illumination (Fig. 4e), consistent with the temporal evolution of cellular forces (Fig. 2).

We next studied the effect of an optogenetic decrease in contractility on tissue deformation. We applied the same experimental protocol described above, but using MDCK monolayers transiently transfected with mito-CIBN-GFP and optoGEF-RhoA. PIV analysis was performed at 40 s time intervals using bright-field images (Fig. 4f–i). As opposed to the case of increasing contractility (Fig. 4a–d), illumination of the central square region of the field of view resulted in systematic cell displacements away from the center of the image (Fig. 4i; Supplementary Movie 9). The time evolution of cell velocity fields showed that tissue expansion lasted longer than compaction (Fig. 4j), consistent with the differences in the time evolution of driving physical forces (Fig. 2). Overall, these results establish that cell contractility and tissue deformation can be controlled with a large dynamic range using optogenetics.

**Optogenetic control of YAP localization**. Contractile forces are well-known to trigger signalling pathways[35]. We thus asked whether our optogenetic tools can be used to control mechanosensitive signalling pathways. To test this possibility, we focused on the transcriptional regulator YAP, which has been extensively shown to translocate from the cytoplasm to the nucleus in response to sustained increased traction forces or substrate stiffness[13,36]. We first studied whether a sustained increase in cell contraction induced YAP translocation. To this aim, we co-transfected MDCK cells with optoGEF-RhoA, iRFP-YAP and CIBN-GFP-CAAX, and subjected them to pulsed illumination. During this process, we monitored the intensity of nuclear iRFP-YAP. As illustrated in the example shown in Fig. 5a,b, optogenetic increase of contractile forces was paralleled by an increase in nuclear YAP (Supplementary Movie 10). Analysis of the distribution of relative changes from baseline after 50 min, revealed an increase of 25% in nuclear YAP (Fig. 5c, green curve). Control cells expressing CRY2-mCherry instead of optoGEF-RhoA subjected to the same illumination protocol did not experience changes in nuclear intensity of YAP (Fig. 5c, black curve). After switching off illumination, nuclear YAP tended to recover baseline levels, indicating reversibility of optogenetic translocation and enabling the possibility of cyclic activation (Fig. 5d). We finally assessed whether relaxation of contractility induced changes in the amount of nuclear YAP. We co-transfected MDCK cells with optoGEF-RhoA, iRFP-YAP and mito-CIBN-GFP and monitored the iRFP-YAP signal in response to pulsed illumination. We observed a reversible decrease in nuclear YAP (Fig. 5e–h, Supplementary Movie 11). Changes in nuclear localization upon illumination were confirmed by immunostaining (Supplementary Fig. 3). Thus, optogenetic contraction and relaxation of cell contractility have opposite effects on nuclear YAP translocation, indicating that optogenetics can be used to modulate mechanosensitive signalling pathways and examine potential transcriptional changes.

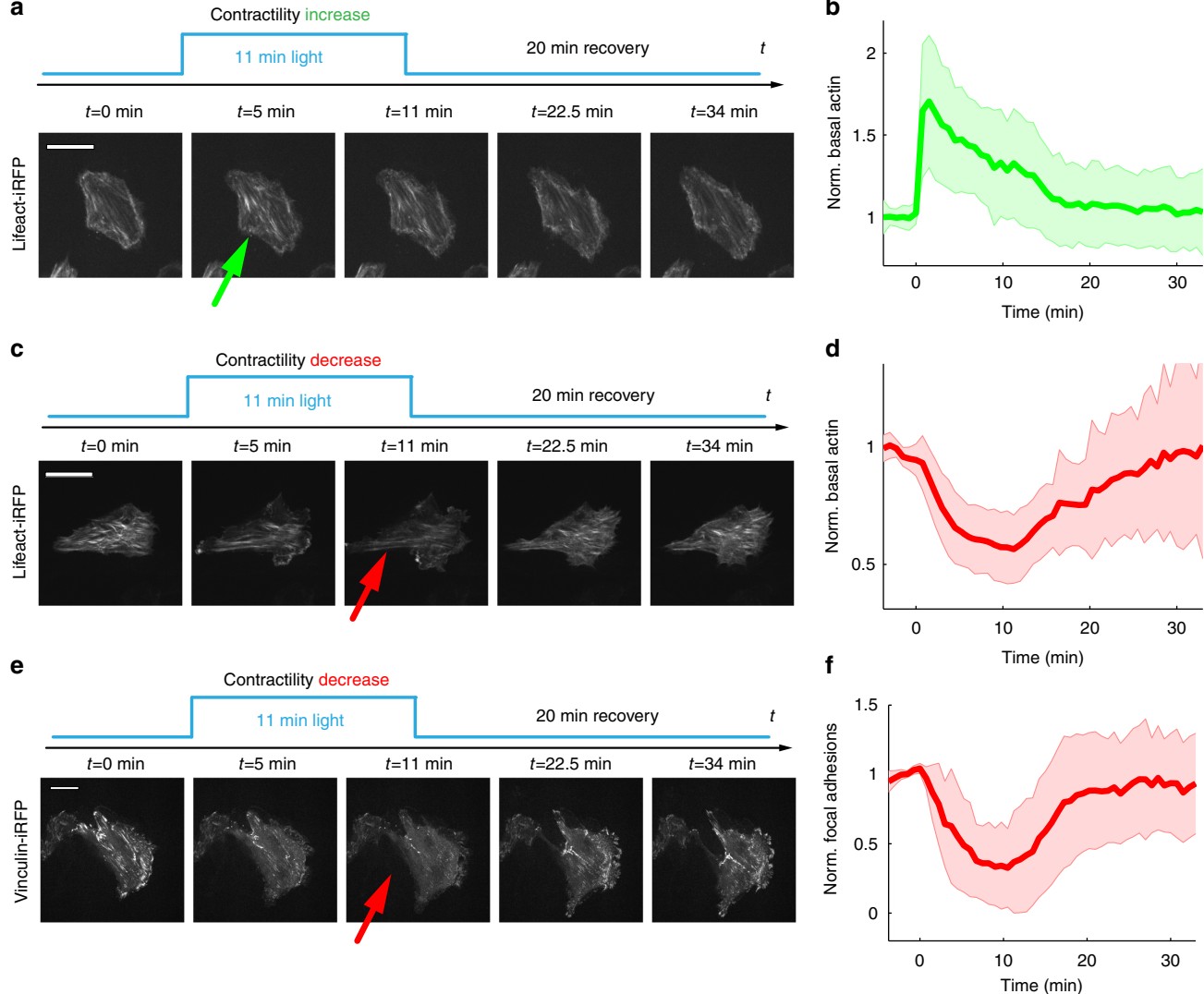

**Figure 3 | Changes in contractility are paralleled by actin and vinculin remodelling.** (**a,c**) iRFP signal of MDCK cells expressing optoGEF-RhoA, lifeact-iRFP and either CIBN-GFP-CAAX (**a**) or mito-CIBN-GFP (**c**) before, during and after 11 min of illumination with blue light. (**b,d**) Quantification of lifeact fluorescence intensity at the basal confocal plane over time normalized by the mean fluorescence intensity before the start of activation (in **b**, $n = 23$ cells; in **d**, $n = 22$ cells). (**e**) iRFP signal of MDCK cells expressing optoGEF-RhoA, vinculin-iRFP and mito-CIBN-GFP before, during and after 11 min of illumination with blue light. (**f**) Quantification of focal adhesion fluorescence intensity at the basal plane normalized by the mean value before activation ($n = 9$). Arrows indicate the time points when largest structural changes occur. Shaded areas are s.d. Scale bars, 20 μm.

## Discussion

We developed and validated two optogenetic tools to control cell and tissue mechanics by acting on the subcellular localization of a RhoA activator. These tools enable to locally increase or decrease cell contractility within tens of seconds. Changes in contractile forces are reversible, paralleled by tissue compaction and expansion, and able to trigger mechanosensitive signalling pathways. To our knowledge, this is the first study that provides a direct measurement of the change in cell–cell and cell–matrix forces in response to optogenetic perturbations as well as changes in signalling downstream from changes in intrinsic cellular mechanics.

Cytoskeleton dynamics has previously been targeted by optogenetic strategies. These strategies include the overexpression of RhoA or a Rho-GEF[26,29], the activation of overexpressed mDia formin[27] or the recruitment of actin disruptors to deplete the actin cortex[28]. These methods enabled changes in cell shape, tissue constriction and cytoskeletal organization. Here we chose to upregulate or downregulate contractility by controlling the subcellular localization of the catalytic domain of the RhoA activator ARHGEF11. Traction forces and cellular tension increased by ∼50% when the activator was localized at the cell membrane, whereas they decreased by approximately fourfold when it was sequestered at mitochondria. Overexpressing a RhoA activator rather than RhoA itself has the advantage that changes in contractility are driven by the endogenous levels of RhoA, thus placing the system close to its physiological tensional homeostasis. Measurements of physical forces demonstrate that our optogenetic approach is versatile and does not compromise cell viability. Indeed, we were able to apply sustained optogenetic activation over periods exceeding 30 min with no appreciable changes in traction forces. Moreover, we were able to apply several cycles of contraction/relaxation with high reproducibility. Our tools can be expressed using standard transient transfection methods or using viral infections to create stable cell lines. Since the amount of light used to control contractility is many-fold lower than the one used for imaging, this system is perfectly adaptable to setups used routinely in live microscopy.

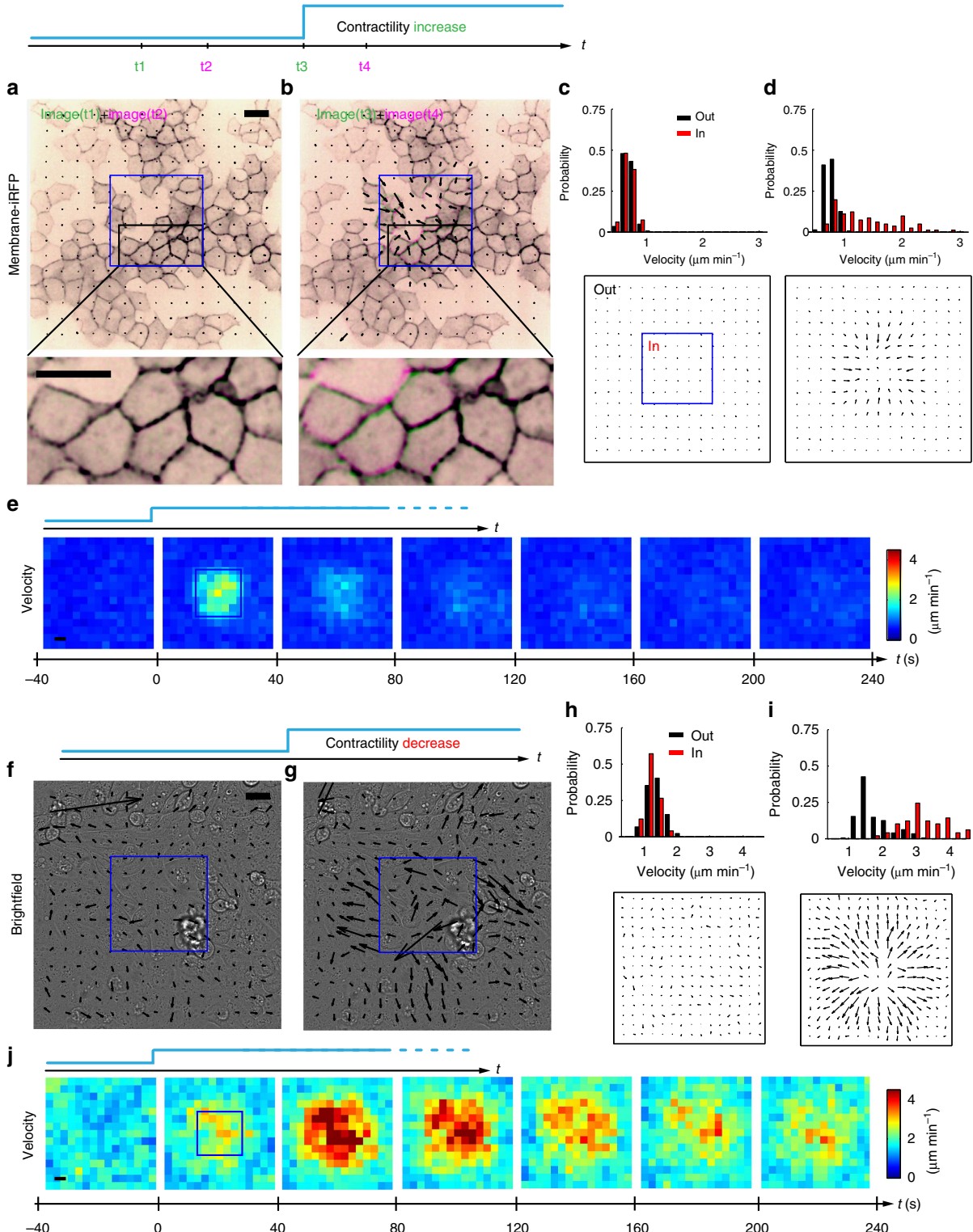

**Figure 4 | Changes in contractility are paralleled by tissue deformation.** (**a,b**) Overlays of two consecutive images of myr-iRFP separated by 40 s (in green first image, in purple second image) before (**a**) and after (**b**) local activation (blue rectangle, one pulse every 20 s for 5 min) of the optoGEF-RhoA/CIBN-GFP-CAAX system. Zooms of the area in the black rectangle are represented at the bottom of the panel. (**c,d**) Histograms of cell edge velocity before (**c**) and just after (**d**) activation. Velocities outside the activation zone are represented in black whereas those in the activation zone are represented in red. Histograms are obtained by averaging 48 velocity fields obtained by PIV analysis. Bottom: representation of the median velocity fields over the 48 experiments. (**e**) Time evolution of membrane velocity magnitude averaged over the 48 experiments. (**f,g**) Bright-field images before (**f**) and after (**g**) local activation (blue rectangle) of the optoGEF-RhoA/mito-CIBN-GFP system. Activation was performed every 40 s for 5 min. (**h,i**) Histograms of velocity fields before (**h**) and after (**i**) the start of contractility relaxation. Velocities outside the activation zone are represented in black and those inside the activation zone are represented in red. Histograms were obtained by averaging 22 velocity fields obtained by PIV analysis on bright-field images. Bottom: representation of the median velocity vectors over the 22 experiments. (**j**) Time evolution of velocity magnitude fields averaged over the 22 experiments. Scale bars, 20 μm.

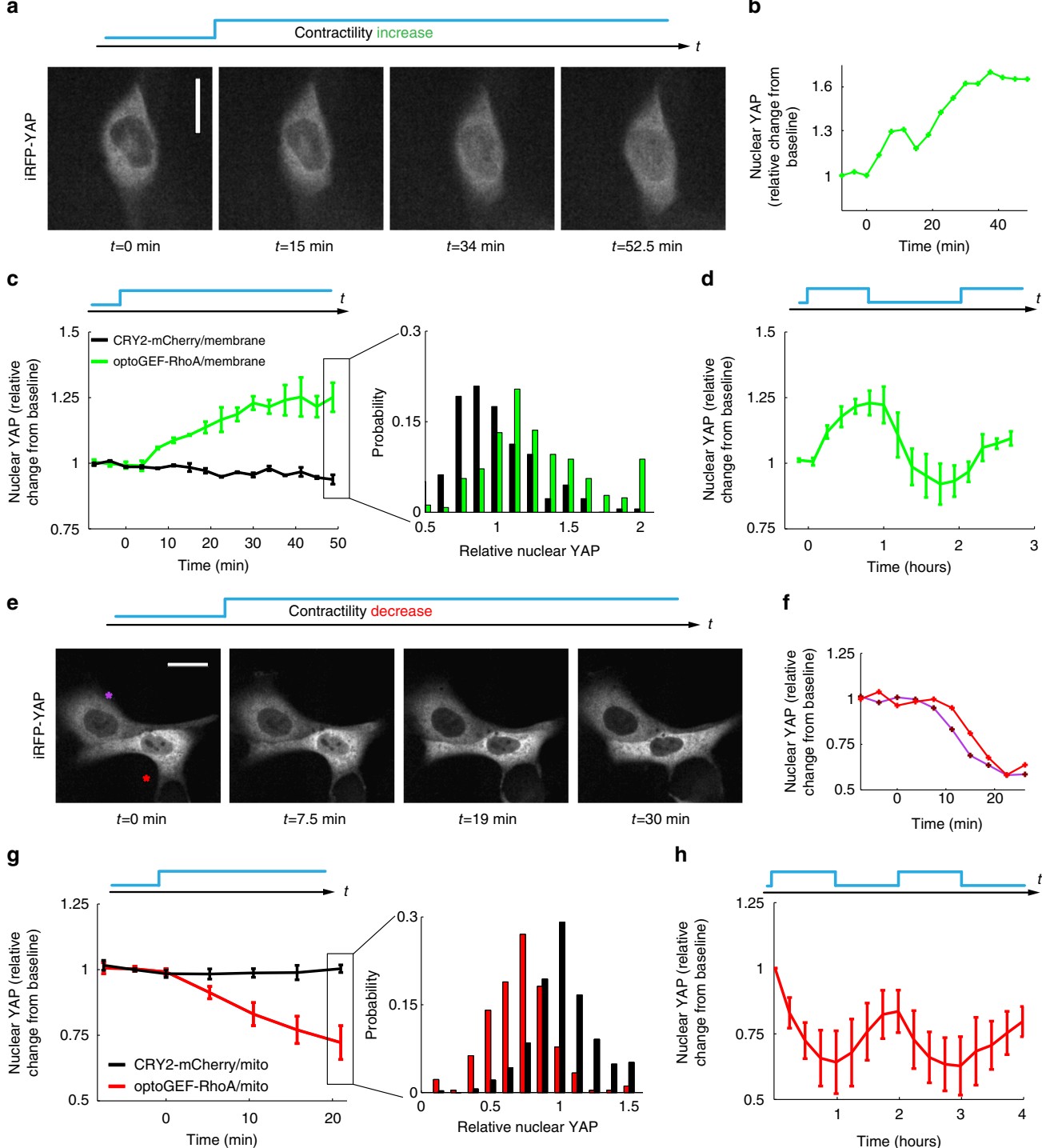

**Figure 5 | Optogenetic changes in cell contractility regulate mechanosensitive signalling pathways.** (**a**) iRFP images of one representative cell expressing optoGEF-RhoA, CIBN-GFP-CAAX and iRFP-YAP before and during blue illumination (7.5 min of imaging followed by 52.5 min of imaging and activation, one activation pulse every 45 s). (**b**) Nuclear YAP fluorescence intensity over time for the example shown in **a**. (**c**) Quantification of relative nuclear YAP over time for control cells expressing CRY2-mcherry, CIBN-GFP-CAAX, and iRFP-YAP (black, $n = 330$ cells) and for cells expressing optoGEF-RhoA, CIBN-GFP-CAAX and iRFP-YAP (green, $n = 270$ cells) subjected to the same activating routine as in **a**. (**c**, right) Distribution of relative nuclear YAP in the experimental population after 50 min of activation. (**d**) Quantification of relative nuclear YAP over time for cells subjected to two activation periods (same period as in **a**) separated by 1h of no illumination ($n = 222$ cells). (**e**) iRFP images of 2 representative cells expressing optoGEF-RhoA, mito-CIBN-GFP and iRFP-YAP during blue light illumination (same illumination protocol as in **a**). (**f**) Nuclear YAP fluorescence intensity over time for the example represented in **e**. (**g**) Quantification of relative nuclear YAP over time for control cells expressing CRY2-mCherry, mito-CIBN-GFP and iRFP-YAP (black, $n = 177$ cells) and for cells expressing optoGEF-RhoA, mito-CIBN-GFP, and iRFP-YAP (red, $n = 250$ cells) subjected to the same activating routine as in **a**. (**g**, right) Distribution of relative nuclear YAP in the experimental population after 20 min of activation. (**h**) Quantification of relative nuclear YAP over time for cells subjected to two activation routines (60 min of illumination, one activation pulse every 60 s) each of them followed by 1h of no illumination ($n = 166$ cells). Error bars are the s.d. between the mean curves of each independent experiment (three independent experiments for each graph). Scale bars, 20 μm.

Besides controlling cellular forces, our optogenetic tools allowed us to control cytoskeleton remodelling and to trigger rapid activation of mechanotransduction pathways. YAP is one key player in the regulation of tissue growth, homeostasis and cancer development[14]. It is regulated by two independent mechanisms. The first one, purely biochemical, involves the Hippo signalling pathway, whereas the second one directly implicates mechanosensitive signalling pathways converting mechanical cues into biochemical signals. Changes in cell spreading area, cell density, substrate rigidity and cytoskeleton contractility have been previously shown to induce translocation of YAP into and away from the nucleus[36]. However, the biophysical mechanisms driving YAP translocation are largely unknown. Here we showed that changes in contractility induce a rapid and reversible modification in the concentration of nuclear YAP (Fig. 5). As such, optoGEF-RhoA provides a new tool to rapidly and locally study the mechanistic relationship between physical forces and YAP localization. We anticipate that it will thus provide a new tool for examining the kinetics of transcriptional changes and protein abundance downstream of mechanotransductory signalling.

The tools reported here have a broad applicability in mechanobiology. At the subcellular level, optogenetic control of cellular forces will enable the study of processes such as cell adhesion, transport and mechanosensing with micrometre resolution and second timescale. At the supracellular level, it will enable researchers to decipher the mechanisms by which epithelial layers deform, remodel and flow. Beyond cell culture systems, the tools developed here should be readily applicable to study morphogenesis in animal embryos. As these tools only require common laboratory equipment and routine genetic manipulations, we expect them to become widely available techniques to control RhoA activity, cellular forces and mechanotransduction.

## Methods

**Cloning.** CRY2-mCherry, CIBN-GFP-CAAX were gifts from Chandra Tucker (Denver, Colorado, United States)[30]. The DHPH domain of ARHGEF11 Guanine Exchange Factor (gene ID 9826, also known as PDZ-RhoGEF) was identified using uniprot.org website. We extended the sequence of interest to retain 8 extra amino acids at each extremity of this catalytic domain following the approach used previously for Intersectin and TIAM1 DHPH GEF domains[23]. The gene was created and inserted into CRY2-mCherry protein by the GenScript company (New Jersey, United states) using Nhe1 and Xho1 cloning sites. Both optoGEF-RhoA and CIBN-GFP-CAAX were inserted into lentiviral backbones (pLVX and pHR') to create stable cell lines.

Myr-iRFP lentiviral vector was a gift from Simon de Beco (Institut Curie, Paris, France). RBD-iRFP, lifeact-iRFP and vinculin-iRFP were gifts from Fahima Faqir (Institut Curie, Paris, France).

iRFP-YAP was cloned by the Protein Expression Core Facility of Institute for Research in Biomedicine (IRB, Barcelona). It was obtained by replacing EGFP from pEGFP-yap-C3-hYAP1 (Addgene, plasmid #17843)[37] by iRFP. iRFP was amplified from the vinculin iRFP plasmid. Cloning was performed using the In-Fusion system; all clones were fully sequenced before use.

Mito-CIBN-GFP was obtained by Gibson assembly cloning, combining the mitochondrial anchor obtained from pcDNA4TO-mito-mcherry-24xGCN4_v1 (ref. 38; addgene plasmid #60913, gift from Xavier Morin, IBENS, Paris, France) with CIBN-GFP amplified from CIBN-GFP-CAAX vector and inserted into peGFP-C1 backbone (Clontech).

**Cell culture.** MDCK-II cell lines (gift from Professor Yasuyuki Fujita, University of Hokkaido, Sapporo, Japan) were cultured with Dulbecco's modified Eagle's medium supplemented with 10% fetal bovine serum, 100 U ml$^{-1}$ of penicillin and 100 µg ml$^{-1}$ of streptomycin. Cells were maintained at 37 °C in a humidified atmosphere with 5% CO$_2$. Cells were tested for mycoplasma and free of contamination. Fluorescent stable cell lines were obtained by viral infection of CIBN-GFP-CAAX and optoGEF-RhoA and then sorted twice within 3 weeks. Transfections were performed using the Neon transfection system (Invitrogen) following the manufacturer's instruction guide.

**Stainings.** Cells were fixed by adding 1 ml of 16% paraformaldehyde for a final concentration of 4% paraformaldehyde (5 min at room temperature). Permeabilization was achieved by incubating with 0.1% Triton X-100 (in PBS) for 5 min at room temperature. Cells were saturated with PBS + 10% FBS (blocking solution) and incubated for 1 h. Primary mouse YAP1(63.7) antibody (Santa Cruz Biot. Catalogue no. sc-101199) was added with corresponding blocking solution at 1:400 dilution and incubated for 1 h. Secondary antibody Alexa fluorophore 647 goat anti-mouse (Thermofisher catalogue no. A-21235) was added at 1:400 dilution and incubated for 2 h.

**TFM gel preparation.** Coverslips were treated with a solution of acetic acid, 3-(Trimethoxysilyl)propyl methacrylate (Sigma), and ethanol (1:1:14) for 15 min, washed three times with ethanol, and air-dried. For 12 kPa hydrogels, a solution containing a concentration of 7.5% acrylamide, 0.16% bisacrylamide, 0.5% ammonium persulphate, 0.05% tetramethylethylenediamine and 4% 200-nm-diameter blue or infrared fluorescence carboxylate-modified beads (Fluospheres, Invitrogen) was prepared in a 10 mM HEPES solution. 18 µl of this solution were immediately placed at the center of glass-bottom dishes and covered with 18 mm diameter glass coverslips. After gel polymerization (1 h at room temperature), the top coverslip was removed. Polyacrylamide hydrogels were incubated with Sulfo-SANPAH under ultraviolet light (5 min). Then gels were rinsed and incubated with 40 ug ml$^{-1}$ of collagen I (Millipore) for 1 h at room temperature and stored overnight at 4 °C.

**Cell imaging and activation.** Cell imaging and activation were exclusively performed using an inverted Nikon microscope with a spinning disk confocal unit (CSU-W1, Yokogawa), Zyla sCMOS camera (Andor, image size 2,048 × 2,048 pixels) and a 60 × objective (NA 1.40, oil). Experiments of Fig. 4,h were done using a 40 × objective (NA 0.95). The set-up was equipped with an incubator to maintain the samples at 37 °C and 5% CO$_2$. Activation of cells located at the center of the field of view was performed by automatically placing a square diaphragm in the main laser path. Activation pulses were 100–200 ms long using a laser at 488 nm with power of ~2 mW (measured at the back focal plan of the objectives).

**Image analysis.** Image analysis was performed with custom-made routines in Matlab (The MathWorks). Image fluorescence analysis of Fig. 1 was carried out by automated segmentation of the GFP signal associated either with the membrane or mitochondria. Images were filtered with a Gaussian filter and then segmented using Matlab 'edge' function. Mean values of mCherry signals were then calculated on the obtained masks. Time lapse intensity was corrected for photobleaching and normalized by its value at the first time point.

Lifeact-iRFP (Fig. 3), iRFP-YAP (Fig. 5), stained YAP (Supplementary Fig. 3) and RBD-iRFP (Supplementary Fig. 1) fluorescent signals were quantified as the mean intensity of square interrogation windows (10–20 µm in side) manually centered at the following locations. For lifeact, center of the cell and background area; for iRFP-YAP, nucleus and background area; for RBD, activated area, non-activated area and background area.

To quantify focal adhesions (Fig. 3), we first filtered fluorescent images of vinculin-iRFP with a Gaussian filter (to remove white noise) and thresholded them to obtain a binary mask. We then summed the intensity of all pixels above the threshold.

**Force measurements.** Traction forces (Fig. 2) were measured using Fourier-transform traction microscopy with finite gel thickness[33,39]. Bead displacements between any experimental time point and its relative reference image obtained after cell trypsinization were computed using home-made particle imaging velocimetry (PIV) analysis. PIV was performed using square interrogation windows of side 128 pixels with an overlap of 0.5.

Tension between and within cells (Fig. 2c,f) was computed using Monolayer Stress Microscopy[34,40]. All maps of tension show the average normal stress.

**Monolayer deformations.** Velocity fields of cell monolayers were measured with a home-made PIV analysis software[41]. Interrogation windows were squares with side 256 pixels and overlap 0.5. Fluorescence images and bright-field images were pre-filtered with a Gaussian filter of 5 pixel width to remove uncorrelated white noise from the image. PIV analysis was applied to images separated by 40 s.

**Code availability.** Matlab analysis procedures can be made available upon request to the corresponding author.

**Data availability.** The data that support the findings of this study are available from the corresponding author on reasonable request. ARHGEF11(DHPH)-CRY2-mCherry and mito-CIBN-GFP plasmids will be available on addgene.

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

## Acknowledgements

We acknowledge M. Coppey, M. Dahan and S. de Beco for their contribution to the initial stage of this project, A. Labernadie and N. Castro for technical assistance, F. Faqir for the production of lifeact-iRFP, vinculin-iRFP and RBD-iRFP, the Institute for Research in Biomedicine (IRB) Barcelona Protein Expression Core Facility for the production of the piRFP-C3-hYAP1 plasmid and P. Roca-Cusachs and all members of the X.T. lab for stimulating discussions. This work was supported by the Spanish Ministry of Economy and Competitiveness (BFU2012-38146 to X.T.), the Generalitat de Catalunya (2014-SGR-927), and the European Research Council (CoG-616480 to X.T., CoG-647186 to G.C.).

## Author contributions

L.V. and X.T. conceived the study and designed the experiments. L.V. performed experiments and data analysis. L.V., G.C., T.W. and A.M.-L. designed and cloned molecular constructs. L.V. and X.T. wrote the manuscript with feedback from all authors. X.T. supervised the project.

## Additional information

**Competing financial interests:** The authors declare no competing financial interests.

**Publisher's note**: 

