## [Peer Review File · Nature Communications]

Reviewers' comments:

Reviewer #1, expert in Rho GTPases (Remarks to the Author):

The article by Valon et al. uses optogenetics to develop a new tool to study contractility. The authors use a DH-PH domain of a RhoGEF that dimerizes in response to blue light to either a plasma membrane or mitochondria targeted binding partner.

The article is thorough and methodic in characterizing the new tool and the results suggest this could be a very useful system to study contractility, and that it can also be adapted to other GEFs to selectively activate other GTPases. As such it should be of interest for the readers of Nature Communications. However, there are some important issues that need to be taken care to strengthen the conclusions of this report.

In Figure 3 the authors use nuclear translocation of GFP-YAP as a readout to measure the effects of changing contractility using the optogenetic DH-PH constructs. There are a couple of concerns with this experiment; First, the authors use GFP-YAP in simultaneously with mito-CIBN-GFP. They claim they can distinguish between the nuclear signal and the mitochondrial signal based on localization. However even in the pictures shown there is mitochondrial signal over the nucleus. Second, the results are not very impressive. In Fig 3b in only one of the three graphs shown the decrease in signal appear large enough. In the third graph (blue line) there is almost no change. Also, the use of the scale is misleading, as different scales are used in each graph to enhance the small changes observed. In 3d, the changes in YAP localization are so modest and the SD is so high that it is hard to understand how they are so significant. Other people have shown that following a decrease in contractility (e.g. blebbistatin treatment), there is a massive (~10 fold) decrease in nuclear YAP (see Dupont et al for example). The results from immunostaining (3e) are somehow better, but I would like to see representative images. Finally, the fact that increasing contractility did not change YAP localization reduces my enthusiasm on these experiments. It is possible that the system is mechanically saturated or the increase on contractility was not enough. However, as is, it leaves the reader wondering whether this is a good tool if it does not respond as expected.

In summary, the initial characterization of the optogenetic tool is good and there is a lot of potential for the construct. However, the experiments testing more physiological relevant conditions are underwhelming and also have some technical problems such as the two GFPs issues.

I recommend the authors to expend this part of the article to convince the reader the construct could be used with confidence in more physiological setup. I could see it being tested with stress fibers or focal adhesions formation for example.

The other thing it would be great to see is a direct measure of RhoA activity (and maybe RhoB and RhoC) to demonstrate that the intended GTPases are being activated.

Reviewer #2, expert in light regulation of signalling pathways (Remarks to the Author):

A. Summary of the key results

The authors developed and validated two optogenetic tools to control cell and tissue mechanics by controlling the subcellular localization of a RhoA activator. Cell contractility can be regulated by light within tens of seconds and changes are reversible, paralleled by tissue compaction and expansion, and able to trigger mechanotransduction pathways.

B. Originality and interest: if not novel, please give references

This is the first study that provides a direct measurement of the change in cell-cell and cell-matrix forces in response to optogenetic perturbations. But a photoactivatable Rac1 has been reported (Wu et al. Nature 461, 104-10. 2009, Wang et al. Nature Cell Biology 12, 591 – 597. 2010), which can regulate RhoA activity in cells. What are the advantages of this new optogenetic approach compared with the previous ones?

The expression of OptoGEF itself increased the system's contractility baseline. So what are the relative levels of contractility of these engineered cells comparing to their non-engineered control cells, which should be more important to understand the mechano-physiology of naturally occurring cells?

C. Data & methodology: validity of approach, quality of data, quality of presentation

In the "Method - Cloning" session, the authors mentioned that eight extra amino acids were added to the DHPH domain. It is not clear on the roles of these extra amino acids. Are these extra amino acids necessary for DHPH function?

The background level in Fig. 3 j is much higher than in Fig. 3 e. What are the possible causes of that?

The contrast/difference shown between Fig. 3f and g is not clear and convincing.

In Fig 2b, area #1(the illuminated area), the response is quite different for different illuminated cells. Is this normal for this system?

In Fig 2 k, there is obvious gradual decrease of the resting state traction level after each cycle. Is this a typical characteristics of the photoactivation system?

D. Appropriate use of statistics and treatment of uncertainties

The manuscript has good practice of appropriate use of statistics and treatment of uncertainties.

E. Conclusions: robustness, validity, reliability

The authors showed they can modulate the cell contractility by light, and the change in contractility is reversible. The results are robust, have been validated from different aspects, and the data is reliable.

F. Suggested improvements: experiments, data for possible revision

In Fig. 4a, the contrast of before/after illumination is not that obvious. Thus, different fluorescent proteins should be used for mitoCIBN and YAP to improve the contrast. The cytoplasm of the cells also became dimmer after illumination, is it due to photo-bleaching? It will be needed to assess the contribution of photo-bleaching to the intensity decrease in the nuclear area.

A direct RhoA assay or RhoA biosensor (Machacek et al. Nature 461, 99-103. 2009) should be needed to show the regulation or activation of RhoA by this method. Contractility is an indirect outcome which may be due to various factors.

The authors should specify the intensity of blue light used for illumination in these experiments and provide the information on how consistent or robust these stimulations can cause on the cellular responses.

Why did the author use a different gene-delivery method for the mitochondria translocation experiment? The expression levels may be quite different for the two different methods. The basal level of the traction force is much higher in the mitochondria translocation experiment than in the cell membrane translocation experiment (Fig.2 g vs. h), which is surprising and warrants a careful and systematic calibration of the expression of synthetic constructs.

To further confirm that optogenetic regulation of cell contractility can activate mechanotransduction pathways, it will be nice if the authors can show the change of downstream gene expression caused by the translocation of YAP protein, which is not that obvious in the current experiments.

G. References: appropriate credit to previous work?

The manuscript has appropriate credit to previous work.

H. Clarity and context: lucidity of abstract/summary, appropriateness of abstract, introduction and conclusions

The manuscript has appropriate abstract, introduction and conclusions.

Responses to reviewers

Reviewer #1, expert in Rho GTPases (Remarks to the Author)

The article by Valon et al. uses optogenetics to develop a new tool to study contractility. The authors use a DH-PH domain of a RhoGEF that dimerizes in response to blue light to either a plasma membrane or mitochondria targeted binding partner. The article is thorough and methodical in characterizing the new tool and the results suggest this could be a very useful system to study contractility, and that it can also be adapted to other GEFs to selectively activate other GTPases. As such it should be of interest for the readers of Nature Communications.

We thank the reviewer for his/her detailed analysis and positive assessment of our work. Our point-by-point response to his/her critique is as follows.

However, there are some important issues that need to be taken care of to strengthen the conclusions of this report.

In Figure 3 the authors use nuclear translocation of GFP-YAP as a readout to measure the effects of changing contractility using the optogenetic DH-PH constructs. There are a couple of concerns with this experiment; First, the authors use GFP-YAP in simultaneously with mito-CIBN-GFP. They claim they can distinguish between the nuclear signal and the mitochondrial signal based on localization. However, even in the pictures shown there is mitochondrial signal over the nucleus.

We agree that the simultaneous use of GFP-YAP and mito-CIBN-GFP was a weakness of our original submission. To overcome this limitation, we generated YAP fused to an iRFP fluorescent reporter. Using this plasmid we confirmed the results of our previous submission showing that YAP translocates from the nucleus to the cytoplasm upon relaxing cell contractility (Fig. 5e-h). We now show, further, that YAP translocates from the cytoplasm to the nucleus upon cell contraction (Fig. 5a-d). One additional advantage of our new probe is that it allows us to monitor YAP localization without activating the optogenetic system.

Second, the results are not very impressive. In Fig 3b in only one of the three graphs shown the decrease in signal appears large enough. In the third graph (blue line) there is almost no change. Also, the use of the scale is misleading, as different scales are used in each graph to enhance the small changes observed. In 3d, the changes in YAP localization are so modest and the SD is so high that it is hard to understand how they are so significant. Other people have shown that following a decrease in contractility (e.g. blebbistatin treatment), there is a massive (~10 fold) decrease in nuclear YAP (see Dupont et al for example). The results from immunostaining (3e) are somehow better, but I would like to see representative images. Finally, the fact that increasing contractility did not change YAP localization reduces my enthusiasm on these experiments. It is possible that the system is mechanically saturated or the increase in contractility was not enough. However, as is, it leaves the reader wondering whether this is a good tool if it does not respond as expected. In summary, the initial characterization of the optogenetic tool is good and there is a lot of potential for the construct. However, the experiments testing more physiologically relevant conditions are underwhelming and also have some technical problems such as the two GFP issues.

We agree that changes in YAP translocation in our first submission were smaller than what might be expected from the literature. We also agree that the lack of translocation in response to increased contractility was somehow disappointing. We have now revisited in full these experiments. First, as mentioned above, we have developed a new iRFP-YAP probe that prevents cross-talk with the mito-GFP signal. Second, we have refined image processing algorithms to perform accurate background subtraction. With these improvements, we now report a ~25% translocation of YAP after 20 minutes of cell relaxation, and > 35% after 45 minutes. These translocation levels are more than double those reported in our initial submission.

With regard to cell contraction, we realized that activation for 20 minutes was not sufficient. Indeed, most studies of YAP translocation report changes occurring over hour time-scales (Benham-Pyle, Science, 2016). By extending the contraction period to 50 minutes, we now report ~25% changes in nuclear levels of YAP. Thus, our new submission now shows YAP translocation upon cell contraction and relaxation.

The reviewer is right in pointing out the high cell-to-cell variability in YAP translocation, which was reflected by large standard deviations in the curves of our original submission. While the median increase or decrease in nuclear YAP after cell contraction or relaxation is ~25-40%, we observed extreme cases exceeding 80%. This variability might originate from endogenous factors such as the initial cell spreading levels or from distinct expression levels of the optogenetic construct. Information about cell-to-cell variability is provided in histograms of Fig. 5c and Fig. 5g. In the curves representing the average increase or decrease in nuclear YAP, we now report the standard deviation between the means of three different days of experiments (Fig. 5c,g). As such, the reader now has access to day-to-day and cell-to-cell variability.

We agree with the reviewer that previous studies have reported decreases by 40-80% using drugs like Y27632 and blebbistatin or exogenous mechanical loading (Piccolo, Nature 2011). The lower levels of YAP translocation in our experiments likely originate from the fact that we are selectively re-localizing RhoA activity without changing its expression levels and without directly perturbing other pathways.

I recommend the authors to expend this part of the article to convince the reader the construct could be used with confidence in more physiological setup. I could see it being tested with stress fibers or focal adhesions formation for example.

We thank the reviewer for this suggestion. We assessed the behaviour of the actin stress fibres and focal adhesions by visualizing temporal changes in lifeact and vinculin (fused to iRFP) after optogenetic contraction or relaxation. Optogenetic relaxation of contractility led to a dramatic subcellular decrease of basal F-actin. The opposite response was observed upon optogenetic contraction. Cell relaxation also induced a pronounced drop in focal adhesions visualized by vinculin-iRFP. By contrast, changes in focal adhesion size were less clear upon cell contraction. All structural changes were fully reversible. We now report this information in new Fig. 3 and new Supplementary Movies 5, 6 and 7.

The other thing it would be great to see is a direct measure of RhoA activity (and maybe RhoB and RhoC) to demonstrate that the intended GTPases are being activated.

We now report direct measurements of RhoA activity using a RhoA biosensor based on the Rhotekin Binding Domain (RBD) fused to iRFP. These experiments are reported in Supplementary Fig. 1 (please see response to part F of reviewer #2). We show an increase in localisation of RBD to regions in which RhoA has been activated.

Reviewer #2, expert in light regulation of signalling pathways (Remarks to the Author):

We thank the reviewer for his/her detailed analysis and positive assessment of our work. Our point by point response to his/her critique is as follows.

A. Summary of the key results

The authors developed and validated two optogenetic tools to control cell and tissue mechanics by controlling the subcellular localization of a RhoA activator. Cell contractility can be regulated by light within tens of seconds and changes are reversible, paralleled by tissue compaction and expansion, and able to trigger mechanotransduction pathways.

B. Originality and interest: if not novel, please give references

This is the first study that provides a direct measurement of the change in cell-cell and cell-matrix forces in response to optogenetic perturbations. But a photoactivatable Rac1 has been reported (Wu et al. Nature 461, 104-10. 2009, Wang et al. Nature Cell Biology 12, 591 – 597. 2010), which can regulate RhoA activity in cells. What are the advantages of this new optogenetic approach compared with the previous ones?

The reviewer is right in pointing out that previous studies have directly tackled the activity of RhoGTPases using optogenetics. In particular, Wu et al. (Nature 461, 104-10. 2009) showed local down-regulation of RhoA activity by locally increasing Rac1 activity. With respect to this previous publication, we believe our approach has two main advantages. First, we aimed at directly controlling RhoA rather than controlling it through Rac1. This provides a more direct control of cell contractility without the side effects that accompany Rac1 upregulation. Indeed, it is highly likely that Rac1 upregulation is not an obligate step for RhoA downregulation. A number of RhoGEFs/RhoGAPs act on RhoA in isolation from Rac1 or act to activate both RhoA and Rac1 with different stoichiometries. Thus, Rac1 and RhoA can be coactivated or activated separately by RhoGEFs. As a consequence, in our opinion, activation of RhoA via RhoGEFs is a more physiological means of controlling activity than indirect control through Rac1. Second, we aimed at controlling cell contractility by modifying the localization of RhoA activity rather than overexpressing RhoA itself. This strategy enables the retention of endogenous RhoA levels and thus places the system closer to its physiological state. We now emphasize the advantages of our approach in the new submission.

The expression of OptoGEF itself increased the system's contractility baseline. So what are the relative levels of contractility of these engineered cells comparing to their non-engineered control cells, which should be more important to understand the mechano-physiology of naturally occurring cells?

The reviewer points out, correctly, that we are over-activating RhoA under baseline conditions due to the over-expression of the catalytic domain of RhoGEF. The comparison between baseline levels in cells expressing optoGEF-RhoA and control cells can be assessed by comparing green curves (contraction) and red curves (relaxation) with black curves (control) in Figures 2g and 2h. Traction values in control are similar to those published in untransfected MDCK cells (see Serra-Picamal et al, Nat Phys, 2012).

C. Data & methodology: validity of approach, quality of data, quality of presentation

In the "Method - Cloning" session, the authors mentioned that eight extra amino acids were added to the DHPH domain. It is not clear on the roles of these extra amino acids. Are these extra amino acids necessary for DHPH function?

We thank the reviewer for highlighting our imprecise wording in this methodological section. The 8 amino acids at each end of the DHPH domain are not "added". They are the 8 amino-acids that flank the DHPH domain in the sequence of full length ARHGEF11. We extended this domain (determined by uniprot.org) to neighbouring amino acids to ensure that the functionality of DHPH

was retained. This strategy had been previously used by Levskaya et al (Nature, 2009) for Intersectin and TIAM1 DHPH GEF domains. We used uniprot.org database to ascertain that these extra domains have no protein function. Given that our constructs were successful at controlling contractility we did not test whether addition of amino acids was critical. We clarified wording in the methods section of our new submission.

The background level in Fig. 3 j is much higher than in Fig. 3 e. What are the possible causes of that?

Our custom-built particle-imaging-velocimetry algorithm measures displacements between consecutive frames using cross-correlation of small interrogation windows. This type of algorithm is highly sensitive to image texture. In Fig. 3e, texture is restricted to the fluorescent probe that labels the membrane. By contrast, in Fig. 3j, texture is determined by the grey levels of a brightfield image, which faithfully captures cell displacements but is affected by local events such as fluctuations of vesicles at the cell surface. These fluctuations are smaller than the signal associated with optogenetic contraction but they result in a higher background noise.

The contrast/difference shown between Fig. 3f and g is not clear and convincing.

We agree that the images obtained with bright field microscopy do not allow the naked eye to clearly discern cell shape and displacements. However, these images enable a reliable characterization of the displacement fields using PIV. In fact, bright field and phase contrast images are extensively used to map cell displacements (see for example, Trepate et al, Nat Phys, 2009; Tambe et al, Nat Mater, 2011). The reliability of the analysis is further supported by histograms and average maps shown in Fig. 4i (compare with control Fig. 4h). Thus, while we agree these images might not be the clearest ones to convey the effect of optogenetic relaxation, they are representative of our experiments. If the reviewer does not strongly object, we would rather keep them as they are. For the sake of clarity we have now increased the weight of the arrows depicting the displacement field in panels 4a,b,f,g.

In Fig 2b, area #1(the illuminated area), the response is quite different for different illuminated cells. Is this normal for this system?

We agree with the reviewer that the different cells of this group are not exerting the same amount of tractions. This could be due to differences in expression levels of optoGEF-RhoA, to differences in endogenous RhoA levels, or to intrinsic mechanical variability in the system. The latter possibility is supported by previous studies showing that cellular tractions within epithelial sheets follow a broad non-Gaussian distribution spanning almost two orders of magnitude (see Trepate et al, Nat Phys 2009). Prompted by the reviewer comment we assessed the variability of traction responses induced by optogenetic activation. Our analysis, shown below for the benefit of the reviewer, reveals that responses are indeed variable but independent of the baseline traction. We note that variability observed here is typical of mechanical measurements in epithelial cells, in which the mean and the standard deviation are of the same order of magnitude (see Tambe et al, Nat Mater, 2011).

Fig R1: Changes in traction induced by optogenetic activation were independent of initial traction.

In Fig 2 k, there is obvious gradual decrease of the resting state traction level after each cycle. Is this a typical characteristic of the photoactivation system?

We agree with the reviewer that there is a gradual decrease of the resting state traction level after each cycle. Prompted by this comment we analysed how general this trend was on a cell-by-cell basis. We found that ~25% of the cells showed a decrease in traction. The origin of this phenomenon is unclear. One possibility is adaptation or remodelling of the actomyosin cytoskeleton over different cycles of activation. The turnover time of actomyosin is on the order of a minute, thus over the duration of a cycle, one might expect the cell to alter its resting morphology. We now mention the attenuation in resting traction but we leave a mechanistic understanding of long-term trends in our system to future studies.

D. Appropriate use of statistics and treatment of uncertainties

The manuscript has good practice of appropriate use of statistics and treatment of uncertainties.

E. Conclusions: robustness, validity, reliability

The authors showed they can modulate the cell contractility by light, and the change in contractility is reversible. The results are robust, have been validated from different aspects, and the data is reliable.

F. Suggested improvements: experiments, data for possible revision

In Fig. 4a, the contrast of before/after illumination is not that obvious. Thus, different fluorescent proteins should be used for mitoCIBN and YAP to improve the contrast.

We agree on this point (see also response to reviewer #1). To uncouple visualization of mitoCIBN and YAP, we have now generated iRFP-YAP which is excited with 647nm light. We have repeated all experiments of YAP quantification using this new construct.

The cytoplasm of the cells also became dimmer after illumination, is it due to photo-bleaching? It will be needed to assess the contribution of photo-bleaching to the intensity decrease in the nuclear area.

We agree that photobleaching has to be carefully ruled out. We do so by showing that the levels of nuclear iRFP-YAP in CRY2-mCherry/mito cells subjected to the exact same illumination pattern than cells expressing optoGEF-RhoA/mito exhibit negligible changes with time (black curves in Fig. 5c,g). We also provide two new supplementary movies illustrating negligible photobleaching despite activation periods of 3-4 hours (Supplementary Movies 10 and 11).

A direct RhoA assay or RhoA biosensor (Machacek et al. Nature 461, 99-103. 2009) should be needed to show the regulation or activation of RhoA by this method. Contractility is an indirect outcome which may be due to various factors.

A great body of literature has established that GEFs are specific to RhoGTPases through their DHPH domains (Rossman, Der, Sondek, Nature reviews, Molecular cell biology, 2005). ARHGEF11 GEF (also known as PDZ-RhoGEF) has been previously shown to interact with RhoA (Zheng et al., BMC structural biology, 2009). Despite this evidence, we agree with both reviewers that the paper would benefit from a direct measurement of RhoA activity after changing optoGEF-RhoA localisation. We note that the optogenetic system is sensitive to blue light and already linked to GFP and mCherry proteins, which precludes the use of FRET biosensors. As an alternative, we resorted to a RhoA biosensor based on the Rhotekin Binding Domain (RBD) fused to iRFP. Sub-cellular activation of those cells showed a local increase of 25% in the RBD signal, which indicates our ability to increase RhoA activity (Supplementary Fig. 1). Control cells expressing CRY2-mCherry instead of optoGEF-RhoA did not show changes in local RBD-iRFP fluorescence intensity upon sub-cellular activation.

The authors should specify the intensity of blue light used for illumination in these experiments and provide the information on how consistent or robust these stimulations can cause on the cellular responses.

We apologise for this missing information, which is now included in the methods section. Pulses of blue light were 100-200ms long using 488nm laser with laser power of ~2mW (measured at the back focal plan of the objectives). As commented above, illumination levels required for activation are lower than those used for imaging and they did not cause significant photobleaching. As regards cell-to-cell and day-to-day variability, this information is provided by standard deviations and histograms in the submission. See also response to reviewer #1 for the specific case of traction variations.

Why did the author use a different gene-delivery method for the mitochondria translocation experiment? The expression levels may be quite different for the two different methods. The basal level of the traction force is much higher in the mitochondria translocation experiment than in the cell membrane translocation experiment (Fig.2 g vs. h), which is surprising and warrants a careful and systematic calibration of the expression of synthetic constructs.

The use of two distinct gene-delivery methods was aimed at showing that our optogenetic system is efficient in stable cell lines but also in transiently transfected cells. As the reviewer points out, differences in delivery methods lead to differences in expression levels, which explain differences in baseline mechanics. Following the reviewer's comment, we now report data using a stable cell line expressing mito-CIBN and optoGEF-RhoA. This cell line shows comparable baseline traction to the CIBN-GFP-CAAX/optoGEF-RhoA cell line used to activate contractility. Mechanical responses to optogenetic activations were qualitatively similar irrespective of the gene delivery method used (Supplementary Fig. 2).

To further confirm that optogenetic regulation of cell contractility can activate mechanotransduction pathways, it will be nice if the authors can show the change of downstream gene expression caused by the translocation of YAP protein, which is not that obvious in the current experiments.

We agree with the reviewer's comment that the manuscript could benefit from additional confirmation of control of mechanosensitive signalling pathways downstream of the mechanical changes induced by optoGEF-RhoA. We now provide such additional information by showing significant YAP translocation into and out of the nucleus in response to contraction and relaxation, respectively. We also show that YAP translocation can be controlled periodically. We appreciate the reviewer suggestion of showing gene expression downstream of YAP translocation. However, we feel that this is beyond the scope of our present study. We prefer to restrict the main body of the present manuscript to time scales shorter than gene expression.

REVIEWERS' COMMENTS:

Reviewer #1 (Remarks to the Author):

The article by Valon et al. uses optogenetics to develop a new tool to study contractility. The authors use a DH-PH domain of a RhoGEF that dimerizes in response to blue light to either the plasma membrane of the mitochondria.

This tool has the potential to be used in a variety of applications to study local responses that involve rapid and reversible changes.

In my initial review, I had some concern on the experiments related to the validation of the optoGEF (e.g. translocation of YAP to the nucleus), as well as some other concerns, mostly technical.

In this revised version, the authors have addressed all my concerns, providing additional data to strengthen the manuscripts in these areas.

In my opinion, the manuscript is now suitable for publication in Nature Communications.

Reviewer #2 (Remarks to the Author):

the authors have provided sufficient information in the revision to address my concerns.